# Compressive Garments in Individuals with Autism and Severe Proprioceptive Dysfunction: A Retrospective Exploratory Case Series

**DOI:** 10.3390/children7070077

**Published:** 2020-07-13

**Authors:** Vincent Guinchat, Elodie Vlamynck, Lautaro Diaz, Coralie Chambon, Justine Pouzenc, Cora Cravero, Carolina Baeza-Velasco, Claude Hamonet, Jean Xavier, David Cohen

**Affiliations:** 1Department of Child and Adolescent Psychiatry, Reference Centre for Rare Psychiatric Diseases, AP-HP, Groupe Hospitalier Pitié-Salpêtrière, Sorbonne Université, 75006 Paris, France; vincent.guinchat@aphp.fr (V.G.); lautaro.diaz@aphp.fr (L.D.); coralie.chambon-devay@aphp.fr (C.C.); justine.pouzenc@aphp.fr (J.P.); cora.cravero@aphp.fr (C.C.); jean.xavier@aphp.fr (J.X.); 2Psychiatric Section of Mental Development, Psychiatric University Clinic, Lausanne University Hospital (CHUV), Prilly, 1011 Lausanne, Switzerland; 3UMR-S 1075 INSERM/UNICAEN, 14000 Caen, France; elodie.vlamynck@gmail.com; 4Interdepartmental Mobile Unit for Complex Situations in Autism, Elan Retrouvé Foundation, 75009 Paris, France; 5Laboratory of Psychopathology and Health Processes (EA 4057), Université Paris Descartes, Sorbonne Paris Cité, 92100 Boulogne-Billancourt, France; carolina.baeza-velasco@parisdescartes.fr; 6INSERM U1061, Neuropsychiatry: Epidemiological and Clinical Research, Department of Emergency Psychiatry and Acute Care, CHU de Montpellier, 34295 Montpellier, France; 7Department of Physical Reeducation, University Paris-Est Créteil, 94000 Créteil, France; pr.hamonet@wanadoo.fr; 8Department of Child and Adolescent Psychiatry, Henri Laborit Hospital Centre, 86000 Poitiers, France; 9CNRS UMR 7295, Cognitive Learning Research Centre, Poitiers University, 86073 Poitiers, France; 10CNRS UMR 7222, Institute for Intelligent Systems and Robotics, Sorbonne Université, 75006 Paris, France

**Keywords:** autism, intellectual disability, challenging behaviors, joint hypermobility, Ehlers–Danlos syndrome, compression garment, proprioceptive dysfunction, pressure therapy

## Abstract

(1) Background: Compression garments (CGs) are an adjuvant treatment for generalized joint hypermobility (GJH), including the Ehlers–Danlos syndrome/hypermobility types. The effects of CGs are likely to be related to better proprioceptive control. We aimed to explore the use of CGs in individuals with autism and severe proprioceptive dysfunction (SPD), including individuals with GJH, to control posture and challenging behaviors. (2) Methods: We retrospectively described 14 patients with autism and SPD, including seven with comorbid GJH, who were hospitalized for major challenging behaviors with remaining behavioral symptomatology after the implementation of multidisciplinary approaches, including medication, treatment of organic comorbidities, and behavioral restructuring. Each patient received a CG to wear for at least 1 h (but most often longer) per day for six weeks. We assessed challenging behaviors in these participants with the Aberrant Behavior Checklist (ABC), sensory integration with the Dunn questionnaire, and postural sway and motor performance using a self-designed motricity path at baseline, two weeks, and six weeks. (3) Results: We observed a significant effect on most ABC rating scores at two weeks, which persisted at six weeks (total score, *p* = 0.004; irritability, *p* = 0.007; hyperactivity, *p* = 0.001; lethargy, *p* = 0.001). Postural control in dorsal and profile positions was significantly improved between before and after wearing the CGs (*p* = 0.006 and 0.007, respectively). Motor performance was also significantly improved. However, we did not observe a significant change in Dunn sensory scores. During the six-week duration, the treatment was generally well-tolerated. A comorbid GJH diagnosis was not associated with a better outcome. (4) Conclusions: CGs appear to be a promising adjuvant treatment for both behavioral and postural impairments in individuals with autism and SPD.

## 1. Introduction

The treatment of challenging behaviors (e.g., self-injurious behaviors, aggression, catatonia, and disruptive behaviors) in children and adolescents with severe autism spectrum disorder (ASD) and intellectual disability (ID) is a complex issue. These behaviors often result in dramatic and even sometimes life-threatening conditions. Many are caused by comorbid medical conditions, either organic or psychiatric [1]. In addition, subjects may experience intense pain from idiosyncratic sources but struggle to localize it and communicate it in an appropriate way, even if the physiological pain response is adequate [2].

Symptomatic treatments include behavioral and family interventions and psychotropic medications, mostly sedative drugs, mood stabilizers, and antipsychotics. To date, only a few atypical antipsychotics have been approved to treat irritability and behavioral impairments associated with ASD or ID [3]. Clinical trials have evaluated an array of therapeutic options providing clinicians with numerous off-label options [4]. In some resistant cases, clozapine [5], intensive behavioral interventions [6], electro-convulsive therapy (ECT) [7,8], or an inpatient stay in a specialized unit using a systematic framework for exploration of comorbid conditions and a multidisciplinary care approach have also been recommended [1]. However, some patients may continue to suffer from residual behavioral disorders. These residual symptoms may be related to alterations in sensory processing, including changes in the integration of information across different sensory modalities (for reviews, see [9,10]). Patients can exhibit distinct sensory processing subtypes associated with adaptive behaviors [11,12], as well as challenging behaviors (for a review, see [13]). Treatments based on sensory-processing difficulties have led to inconclusive results (for meta-analyses, see [14,15,16]).

In the present study, we focused on proprioceptive processing. Proprioception is the sensory registration of the ongoing spatial configuration of the body, which includes the position of body segments in space, the force and speed of movement, and the integration of gravity and body balance. Proprioception is therefore known to impact behavioral regulation and motor control [17]. The role of proprioception is controversial in ASD. Some authors have postulated that patients with ASD have a primary proprioceptive deficit causing an over-reliance on visual information [18] or dysfunction during pointing tasks [19]. An opposite hypothesis postulated that patients with ASD do not have a primary proprioceptive deficit [20] and may be more proprioceptive learners, as they construct internal models of actions which rely on proprioception to a greater extent than normal subjects, in order to compensate for other sensory particularities [21,22,23,24,25]. In the context of severe ASD and ID, many factors may independently impede proprioceptive inputs and integration, such as chronic pain, seizure, fever, or dystonia [26,27,28]. Therefore, in both perspectives, proprioceptive processing abnormalities could be involved in motor control abnormalities and postural sway [10], two acknowledged comorbidities of severe ASD and ID [29,30]. Some maladaptive behaviors associated with weak proprioceptive integration are identifiable in clinical practice through observations: postural maladjustments (e.g., tonic postural collapse with unusual points of gravitational support, such as the hands or the head), sensory-motor stimulations (e.g., twirling), and unsteady balance [31,32,33,34]. These processing abnormalities may underlie some challenging behaviors. If so, an increase in or homogenization of the proprioceptive input would likely benefit patients with autism.

Physical therapies (e.g., deep pressure therapy or Qiqong therapy) or orthesic treatments (e.g., clothes marketed for ASD, weighted jackets or blankets, and snug vests) have been designed to address these proprioceptive processing abnormalities, but empirical data supporting their use are scarce [35,36,37,38,39,40]. Here, we aimed to study the effect of whole-body compression garments (CG), a medical device currently used to reduce proprioceptive dysfunction in Ehlers–Danlos syndrome (EDS). EDS is a heterogeneous group of hereditary connective tissue diseases leading to systemic damages. Among others, patients experience chronic fatigue, musculoskeletal pain, proprioceptive impairment, clumsiness, and postural problems [41,42]. The EDS revised criteria distinguish 13 subtypes, the hypermobility subtype (hEDS) being the most frequent [43]. Joint hypermobility and cutaneous hyper-elasticity are the two main criteria for hEDS. Recently, it has been associated with ASD [44,45,46]. The hEDS/ASD comorbidity could account for a deficit of proprioceptive inputs. Amongst several procedures, CGs have been specifically designed for patients with EDS, in order to relieve their pain and fatigue and improve their mobility [47]. CGs may exert a homogenous mechanical effect enhancing joint coaptation, increasing the pressure of the subcutaneous connective tissue to a normal range and improving the somatosensory/proprioceptive feedback to the brain [48].

The exact relationship between ASD and EDS is unknown. However, ASD has two characteristics in common with hEDS: (1) frequent pain experience, which is difficult to recognize by health professionals; and (2) the presence of motor maladjustment and proprioceptive dysfunction. EDS is included in the broader spectrum of generalized joint hypermobility (GJH), for which proprioceptive deficit has also been reported (for review, see [49,50]). In addition, GJH is present in many neuro-developmental syndromes, such as XFra or Down syndrome, which are associated with ASD and ID [51,52]. we aimed to retrospectively explore whether CGs can improve postural control, gross motor skills, and challenging behaviors in individuals with ASD, residual challenging behaviors, and severe proprioceptive dysfunction (SPD) because: (1) CGs are an adjuvant treatment for hEDS that induce better proprioceptive control; (2) some individuals with ASD may show severe sensory proprioceptive dysfunction; and (3) gross motor control, posture, and some challenging behaviors in subjects with autism may mirror proprioceptive dysfunction,.

## 2. Materials and Methods

### 2.1. Study Design

This study was comprised of a retrospective single-site, open-label case series. All patients were recruited from the neurobehavioral inpatient unit of the Pitié-Salpêtrière University Hospital, a third-line unit dedicated to the management of severe challenging behaviors in individuals with ASD and ID [1]. It is articulated with three mobile units, and its intervention covers the entire territory of the Ile de France region (12 million inhabitants).

### 2.2. Participants

All participants were treated in strict compliance with the Declaration of Helsinki. CGs are currently funded by the French healthcare system in the indication of EDS. Patients with both ASD and EDS who present severe challenging behaviors may officially benefit from CGs. Other patients were able to obtain CGs as a gracious gift from Novatex Medical for the SPD indication. As we employed a retrospective design for this case series, no external review board was required [53]. However, given the compassionate nature of the intervention, written informed consent was obtained from the parents of all participants when CGs were proposed. In addition, as requested by French regulation rules, all patients/families were contacted for inclusion of the data in a retrospective study; in particular, the database was approved by the *Commission Nationale Informatique et Liberté* (CNIL or National Informatics and Freedom Commission) under the number No. 1256552.

Since 2015, we have proposed CGs for patients when: (1) a current diagnosis of severe ASD and ID was confirmed by a specialized clinical assessment (DSM-5 criteria) [54]; (2) patients presented severe behavioral disturbances, such as aggressive behaviors, self-injurious behaviors, severe motor hyperactivity, severe stereotypies, or catatonia; (3) patients presented residual and impairing symptoms after a three-month period of an intensive and multidisciplinary care approach, where resistance to treatment was evidenced by a low level of improvement using the Aberrant Behavior Checklist (ABC) [55], which is routinely used in the unit to monitor challenging behaviors; and (4) patients presented SPD with or without EDS that had persisted at the time CGs were proposed.

Autism symptom dimensions were assessed using the current Autism Diagnostic Interview-Revised (ADI-R) addressed to parents [56]. Multidisciplinary management was initially conducted in the inpatient unit. All patients were referred to dental, gastroenterology, neurology, ENT, and eye specialists. The following systematic examinations were performed: blood tests, including genetic (array comparative genomic hybridization, aCGH) and metabolic tests; electroencephalography; brain MRI; and upper gastrointestinal endoscopy. Extensive psychiatric management was employed, including the use of ECT in some patients. In addition, a specialized comprehensive psycho-educational program, mainly based on Applied Behavior Analysis (ABA) principles, was implemented in the inpatient unit as part of the management strategy (decision guidelines are described in [1]).

SPD was assessed using a qualitative assessment performed by two occupational therapists during each individual’s routine examination. We extracted the items describing behaviors that were possibly related to proprioception from the existing clinical scales [34,57,58]. These behaviors were described as symptomatic if they led to permanent and severe impairment. Then, we recorded information about 4 dimensions of symptoms and 19 sub-dimensions: (1) tone (hypotonia, hypertonia, and tonic fluctuation); (2) postural control (tiptoeing, pushing others or objects, leaning on the ground or others, head down, searching for pressure or enjoy being pulled, and balance difficulties); (3) motor control (crashing and falling, movement overactivity, poor joint alignment, stiff movements, and movement avoidance); and (4) stereotypies (seeking body vibrations, turning one’s head, spinning and running, swinging movements, breath-hold, and shouts). Given the lack of a consensus definition and the fact that some sub-dimensions may be related to other types of neurological dysfunction (e.g., cerebellar or vestibular dysfunction), we diagnosed patients with SPD if symptoms were present in at least two dimensions (Table 1).

The use of prescribed antiepileptic and psychotropic medications must have been stable for at least four weeks. Patients were systematically screened for EDS and GJH criteria by an expert physician. We offered the use of CGs to 14 patients. Six patients started the protocol during their inpatient stay and eight after discharge from the hospital.

### 2.3. Intervention: The Compression Garments

The CGs used in this study were customized based on the needs of each patient by orthotic and prosthetic practitioners (Novatex Medical©). Based on 36 body measurements, CGs included tailor-made pants, vests, and mittens, which covered the entire body of all participants (i.e., trunk and upper and lower limbs; Figure 1). Closure systems varied, according to the behavior of the patients. The vest was attached to the pants by a zipper located at the waist for some patients, and the vest closed in front and/or behind with a zipper. In some clothes, we added zippers at the ends of long sleeves, as well as scapular elastic recoil or removable protective foams applied for the purpose of protection (i.e., biting hands).

The technical specifications of proprioception clothing were the same as those of CGs for large burns, but the exerted pressure was lower. For a perimeter of 24 cm, the pressure ranged 10–13.5 mmHg. For a perimeter of 55 cm, the pressure ranged 6–10 mmHg. The fabric was constructed by chain knitting with the insertion of threads in the weft direction. This method increases the elasticity and multidirectional resistance for better resistance to fatigue. Its composition was 59% polyamide and 41% elastane, and the weight per area was 250 g/m^2^. Chain length extension and elongation direction frame both exceeded 100%. The UV protection level was 50+ and the fabric had an Öeko-tex-Standard 100 class II certification. Each patient received a CG orthosis to wear for at least 1 h during the day for six weeks. This minimum threshold was justified by the problems of enuresis, encopresis, and undressing experienced by some patients.

### 2.4. Clinical Measures

The primary outcome assessed at enrolment and six weeks was postural imbalance with and without CG. We could not measure postural sway using a motorized force platform, as it required too much compliance from our patients. We used a basic, non-instrumented postural test with the vertical of Barré (via a laser) and a blue plank on the floor (see Figure 2) to position the two feet of the patient. We asked patients to stand still while barefoot with arms hanging freely, feet positioned with a Fick angle at 30°, and to focus on a visual reference mark fixed 1.5 m in front of them. We performed a clinical postural analysis, according to Vallier’s classification, in order to identify the type of imbalance affecting the subject [59]. At least two trained orthesists performed the measurements. Several pictures were taken when the posture was considered stable. Postural sway parameters included the anteroposterior (profile) and mediolateral (frontal) sway standard deviations (SD-profile/SD-frontal; mm). We used two approaches to assess changes: the difference between post- and pre-intervention profiles and frontal deviation, and a Clinical Global Impression Likert scale (worse, unchanged, improved, and substantially improved) based on clinician observations during the test. Figure 2 shows the procedure applied to a catatonic patient. Written informed consent was obtained from the patient’s parents for the publication of these images.

We assessed sensory motricity with two variables at enrolment and six weeks: (1) the Dunn sensory profile to assess sensory integration, which produces 22 sub-scale scores (i.e., auditory processing, visual processing, vestibular processing, touch processing, multisensory processing, oral sensory processing, sensory processing related to endurance/tone, modulation related to body position and movement, modulation of movement affecting activity level, modulation of sensory input affecting emotional responses and activity level, modulation of visual input affecting emotional responses and activity level, emotional/social responses, behavioral outcomes of sensory processing, sensory seeking, emotionally reactive, low endurance/tone, oral sensory sensitivity, inattention/distractibility, poor registration, sensory sensitivity, sedentary, and fine motor/perceptual) and a total score [57]; and (2) a self-designed motricity path to assess motor performance (Figure 3). Two occupational therapists adapted the simplest tasks from standardized tools, such as the Movement Assessment Battery for Children, 2nd edition (MABC-2) [60] to assess gross mobility. The scenario included ten workshops requiring general dynamic co-ordination (jumping, walking, and balance). The evaluation was based on a review of the video and of the fulfillment of a certain number of qualitative observation criteria, defined according to three variables: “yes”, “no”, and “emerging”. Fifty items were assessed. This course was performed on Day 0 without CGs and after six weeks of intervention with CGs. The sequences of the tasks were not repeated, in order to avoid motor learning between sessions. All tasks were performed in the same room and by the same professional team at the child psychiatry department of the Pitié-Salpêtrière Hospital. When CGs were indicated after discharge, a time-lapse was respected such that the patient could adapt to his/her non-hospital environment.

Finally, ABC [54] was monitored at baseline, two weeks, and six weeks, in order to assess behavioral changes. It included scores for several subscales, irritability, hyperactivity, lethargy, inappropriate speech, and stereotypic behaviors, as well as the total score. A clinical expert checked that the behavioral variables were stable at least two weeks prior to enrolment.

### 2.5. Statistical Analysis

Statistical analyses were performed using the R Software, version 2.12.2. For all tests, the level of significance (alpha) was fixed at 5%. Categorical variables are reported as frequencies and percentages. Continuous variables are presented as means (standard deviations). Given the limited sample size, we used non-parametric tests: the Friedman test (a non-parametric equivalent of repeated measures ANOVA) to assess changes in ABC scores and the Wilcoxon paired signed rank test to compare pre- and post-intervention scores for other variables (Dunn sensory profile, sway SD-profile and SD-frontal measures, successful completion of the motricity path, and emerging and failed item scores).

## 3. Results

### 3.1. Characteristics of the Participants and Adverse Events

The study sample included 14 participants: 13 males and 1 female. Our cohort consisted of seven children and adolescents (aged 8–15 years) and seven young adults (aged 18–30 years). All of them had severe autism, no functional language, and a mean developmental age of less than 36 months with respect to the Vineland Adaptive Behavior Scale II (VABS-II). Prior to enrollment, four patients had experienced life-threatening complication of their behavioral problems (severe malnutrition, N = 2; post-traumatic sepsis, N = 1; and uncontrolled seizures mostly triggered by stereotypic hyperventilation, N = 1) and three others were initially referred after a few years of seclusion or mechanical restraint, mainly in a psychiatric unit. Table 2 summarizes the sociodemographic and clinical characteristics at baseline, including the main reasons for referral, duration of hospitalization, current medications, VABS-II developmental age [61], and ABC total scores. All patients were receiving medication at baseline. Treatment at baseline and eventual changes are summarized in Table 2. All antipsychotic dosages are presented in mg of the equivalent chlorpromazine dose (EqChlor). The most frequently used medication was aripiprazole, but most patients receiving drug treatments were taking multiple medications that included off-label drugs (e.g., clozapine). For all but two of the patients, prescriptions remained stable during the study. These patients continued to use the same drugs, but minor changes in the respective dosages occurred: one received an additional 150 mg EqChlor of antipsychotics and the other received a dose reduced by 105 mg EqChlor. There were no other changes in patient management during the protocol. Appendix A details the EDS criteria for each participant. Seven participants (50%) presented positive criteria for hEDS or HSD [43].

Although all patients were initially hospitalized, only six of them started and finished the protocol in the ward. The other patients started the protocol after discharge, either in the specialized institutions where they were treated (N = 4) or at home (N = 5, with the support of professional caregivers and motivated parents). To limit this issue, when a discharge was programmed, professionals of our team were affected to the original facilities to ease the transition and starting of the protocol when patients became stable in their environment. Caregivers were then called twice a week by L.D., a psychologist, and asked to report the duration that their children spent wearing CGs, record any incidents, and assess the challenging behaviors. We thus ensured adherence to the protocol.

Only one adverse event must be noted: one patient, who was generally reluctant to take off the CG, was accidently allowed to sleep with the CG on all night long. The next morning, the sleeve of the CG had rolled up and the patient presented a slight and transient hand edema. No side effects were noted for those patients who strictly followed the protocol. One patient wore the CG approximately 2 h a day, with substantial beneficial effects on behavior; six patients wore it between 4 and 8 h a day; and the others from the time they woke to bedtime. Among these individuals, three would have preferred to continue wearing the CG even while sleeping, which was not allowed.

### 3.2. Outcome Variables

Table 3 shows the clinical changes (delta and effect size) in outcome variables between the baseline and six-week outcome. We observed a significant effect on most ABC rating scores; on postural control in dorsal and profile positions; and on motor performance. However, we did not observe a significant change in Dunn sensory scores. At an individual level, we found an improvement of challenging behaviors for 12 (85%) patients after six weeks. For two (15%) patients, challenging behaviors worsened. Appendix A shows the changes in ABC total scores and sub-scores between baseline and T1 (two weeks) and T2 (six weeks). We observed a significant decrease in the ABC total scores (*p* = 0.004), as well as ABC-irritability (*p* = 0.007), ABC-hyperactivity (*p* = 0.001), and ABC-lethargy (*p* = 0.001) sub-scores. Significant changes in scores for the ABC-stereotypies and ABC-inappropriate speech sub-scales were not observed. According to the results of the post-hoc analysis, significant changes occurred between baseline and two weeks, where the clinical improvement persisted at six weeks with no supplemental improvement between two and six weeks. As a preliminary analysis, we also explored whether joint hypermobility moderated the observed improvements. Appendix A shows the ABC total score per individual, according to the joint hypermobility comorbidity. A significant moderating effect (*p* = 0.096) of joint hypermobility on the ABC total score was not observed.

Table 4 describes the postural imbalance observed using the vertical of Barré. Of 12 patients who could accept the testing conditions, none had baseline normal posture. In dorsal view, 11 patients had a lateral deviation which disappeared with CGs in three (25%) patients and was reduced in five others (45%). Eleven patients could be assessed in profile view. Nine had an anterior or posterior scapular deviation. Using CG, seven patients dramatically improved; five (55%) normalized and two (22%) clinically improved. Taken together, each one of 12 patients that could be assessed showed a postural improvement in at least one view, and three patients in both views.

## 4. Discussion

### 4.1. Summary of the Results

CGs are therapeutic devices which have shown potential for improving the well-being of patients with severe autism and proprioceptive disturbances. In this retrospective study, we identified effects on postural sway, motor performance, and challenging behaviors at six weeks. Given our theoretical background, we explored whether patients with ASD and HSD had a specific clinical response; however, a more pronounced effect on the hypermobile sub-group of patients was not observed (Appendix A). A fundamental question to explore is whether this lack of a pronounced effect was due to the small size of our sample or whether it indicates an independent effect of CGs (e.g., on proprioceptive disturbance in autism itself or by means of providing constant and consistent stimulation to the pressure sensors in the skin, which enables the patient to ignore more disturbing unpredictable sensations through this predictable and constant “noise”). However, given the exploratory nature of the study, we should remain cautious before drawing any generalization. Based on our clinical experience, we discuss some specific issues that require more research.

### 4.2. Problems Raised by the Use of CGs in Patients with Autism

Among the wide variety of therapeutic modalities, compression therapy is a recurring treatment with a long history. First, in autobiographical accounts, several individuals with high functioning autism have reported the need for compression [9,62]. For example, Temple Grandin described how she built a “hug machine” when she felt stressed [63]. Second, practitioners often observe low-functioning patients with autism spontaneously wrapping themselves in sheets [64]. Third, therapeutic procedures aiming to facilitate sensory integration, such as therapeutic body wraps (TBW), have been proposed with encouraging preliminary results [65,66]. However, they remain controversial [67]. In addition to definitive efficacy, the use of CGs raises two other issues.

On the one hand, how a CG works is challenging to describe. Therefore, the theoretical background justifying the use of CG in our population is speculative and based on: (1) the frequent co-occurrences between ASD and proprioceptive dysfunction [29,30,68] and ASD and joint hypermobility [44,45,46]; and (2) the existing clinical use of CGs in EDS [47,48]. Hypotheses regarding how CGs work include: (1) a better sensing of the limbs in space by stretching the skin. From this perspective, the stimulation of cutaneous mechanoreceptors enhances tactile input, alters the excitability of the central nervous system, and modulates proprioceptive afferent feedback loops [69]. These changes, in turn, facilitate motoneuron control and attenuate muscle oscillations during movement [70]; (2) stabilization of muscle architecture [71] and muscle activity [72]; (3) improvement of movement sensation, joint positioning [73], and proprioceptive acuity [74,75,76,77]; and (4) relief of muscle fatigue [78]. CGs have become increasingly popular among athletes who seek to enhance their performance, reduce fatigue, and decrease injury risks during training and competition [79,80], or to facilitate recovery [81]. In therapeutic medicine, both antalgic and sensory effects are also targeted; for example, post-burn injury syndrome [82], stabilization of weak joints [83], and substitution for proprioceptive deficits in patients with Parkinson’s disease [84].

On the other hand, there are also ethical concerns. The CG applied in this case series had several advantages over the usual orthesic treatments (i.e., clothes marketed for autism, weighted jackets or blankets, and snug vests) and TBW. First, as previously mentioned, the CG has already been used in various medical fields (e.g., EDS, sport, and large burns). Second, it offers a better appraisal of the compression effect, due to the mechanical properties of the CG material and the fit of the material relative to a patient’s anthropometry and morphology. Third, the procedure is simple, not constraining, and allows total freedom of movement (as opposed to TBW). The CG is easy to use and maintain, and it is customized according to each patient’s ergonomic needs. Patients with EDS have attested to its good tolerability [85]. In our series, the tolerance and acceptability of the device was excellent. Finally, the use of a CG seemed an acceptable ethical alternative to the existing procedures for our population. We focused on patients with severe autism and resistant challenging behaviors for which therapeutic options are dramatically lacking, often requiring immediate measures of seclusion and leading to insoluble institutional care [1].

### 4.3. Limitations

The study has several important limitations: First, the clinical sample was biased towards the most severe and resistant cases. Second, the sample was heterogeneous, as proprioceptive dysfunction was associated or not with joint hypermobility or hEDS. Third, the sample size was limited and the study was retrospective and exploratory with no historical control group and no blind clinical assessments. Fourth, due to the prevalence of challenging behaviors, assessments were tailored to the individual skills of patients. While many proprioception tests are available in the literature [34,86], none were suitable for our population. We had to design a qualitative scale based on the most common symptoms, in order to include patients with proprioceptive disorders. This qualitative scale has not been validated. However, a new scale for assessing motor skills and proprioception has recently been developed and validated [87]. We hope to use this new tool, in the future, to address this limitation. Postural analyses were not recorded on a force platform [88,89]. Furthermore, standardized motor tests (e.g., MAB-C) were too demanding for the patients enrolled in our study, in terms of participation, attention, and verbal communication [60]. Elaborating a self-designed motor path was our only option to assess gross motor skills; however, this approach did not adequately resolve the limitations with compliance and lacked reliability. Finally, the last limitation to mention is the great variability in the duration for which the CG was worn. Our recommendation to wear CGs at least 1 h during the day was designed to include a population with a very low level of compliance. Nevertheless, our minimum threshold was much longer than the duration of most physical therapies, as only one patient wore the CG for less than 4 h a day (with substantial benefits). The limitation of sample size precluded a sub-group analysis based on the length of time the patient wore the CG.

### 4.4. Implications for Future Research

This pilot retrospective case series is only exploratory in nature. It indicates the potential value of CGs for improving challenging behaviors, posture, and gross motor skills in individuals with severe autism and proprioceptive dysfunction over a six-week period. The four patients in our sample who experienced the most striking effects of the device continued to wear it daily for several months after the protocol was complete. These promising results encourage the design of a randomized and controlled multi-center protocol to assess not only sensory-motor skills but also the core symptoms of autism. Alternative questions refer to: (1) whether or not we should propose CGs only for patients with challenging behaviors; and (2) whether or not we should also explore the use of CGs over longer periods of time. Indeed as sensory motor dysfunctions affect children’s daily living skills [90,91,92] and reduce their compliance with educational programs, a longitudinal study may determine whether CGs facilitate their implementation and improve their outcomes on daily skills.

Finally, some future research should focus on how the CG works. In our clinical sample of complex patients, we were unable to determine whether the CG modulated basic functioning, as we found concomitant improvements in gross motor skills, postural sway, and challenging behaviors. These findings may mirror the observation that the CG might contribute to the acquisition of better motor patterns, either by reducing a primary deficit or strengthening the input in people who rely on proprioception for motor learning [93,94], leading to better behavioral outcomes [95]. Furthermore, we do not know if CGs work only as a sensorial substitution or if they also contribute to an extensive modification of internal models of action. The definition of a period of persistent effect once the patient has removed the CG would represent substantial progress.

## 5. Conclusions

CGs appear to be an innovative and interesting adjuvant treatment for both behavioral and motor impairments after multidimensional and integrative proper care in individuals with severe ASD, ID, and comorbid SPD. These promising results require confirmation by further evidence-based research, which should consider not only sensory-motor skills but also the core symptoms of autism.

## Figures and Tables

**Figure 1 children-07-00077-f001:**
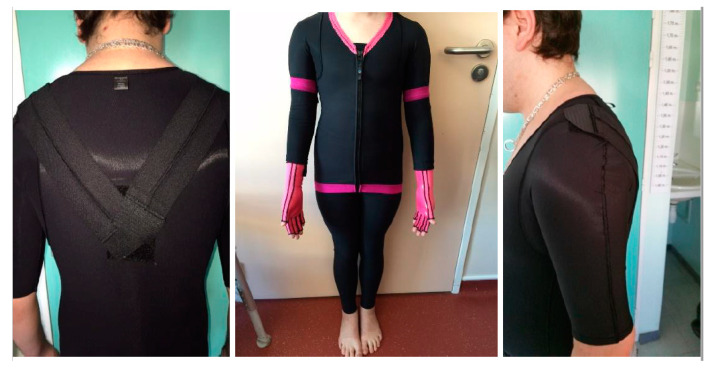
Examples of compressive garments (Novatex Medical).

**Figure 2 children-07-00077-f002:**
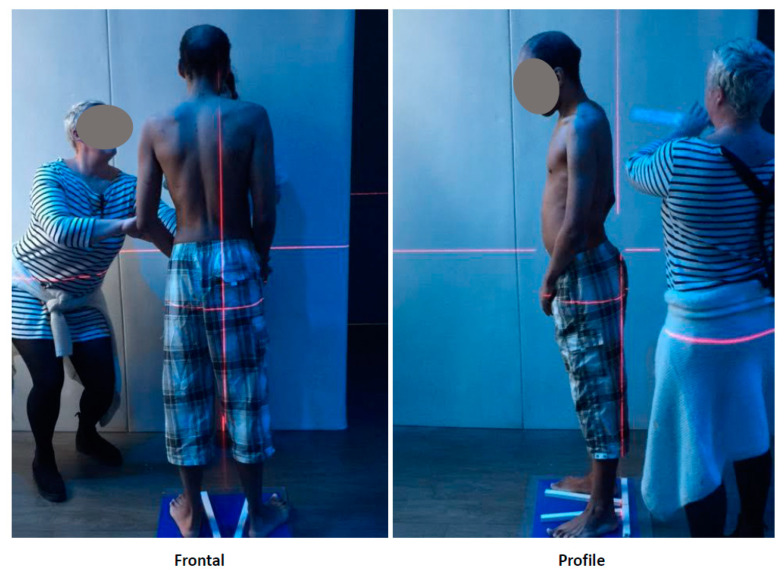
Measurement of postural sway parameters using Vallier’s classification: (**Left**) mediolateral (frontal); and (**Right**) anteroposterior (profile).

**Figure 3 children-07-00077-f003:**
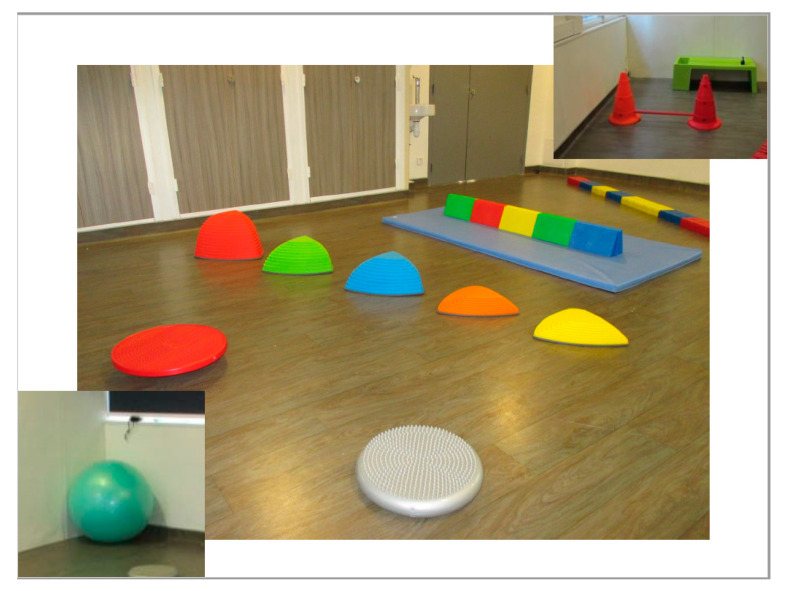
Setting of the self-design motricity path.

**Table 1 children-07-00077-t001:** Clinical proprioceptive characteristics at admission of the 14 participants who completed the protocol stratified according to four dimensions that included a total of 19 symptoms.

	Tone (max = 3) (Hypotonia, Hypertonia, and Tonic Fluctuation)	Postural Control (max = 6) (Tiptoeing, Pushing Others or Objects, Leaning on the Ground or Others, Searching for Pressure or Enjoy Being Pulled, Head Down, and Balance Difficulties)	Motor Control (max = 5) (Crashing and Falling, Movement Overactivity, Poor Joint Alignment, Stiff Movements, and Movement Avoidance)	Stereotypies (max = 5) (Spinning and Running, Seeking Body Vibrations, Swinging Movements, Turning One’s Head, Breath-Hold, and Shouts)
P1	1	4	4	3
P2	2	3	3	3
P3	2	4	1	3
P4	2	2	4	2
P5	1	2	2	0
P6	1	3	1	4
P7	3	5	3	3
P8	1	3	1	1
P9	0	4	1	5
P10	2	3	2	2
P11	2	1	2	0
P12	1	2	2	3
P13	2	2	3	2
P14	1	3	4	3

The presence of one symptom in each dimension receives a score of 1 point. The total score for each dimension represents the sum of the symptom scores. max: maximum.

**Table 2 children-07-00077-t002:** Patient socio-demographic and clinical characteristics at baseline.

Socio-Demographics	
Males: N (%)	13 (93)
Age (years): mean (±SD) [range]	18.2 (±5.5) [8.2–30.0]
SES: N (%), low/middle/high	4 (29)/3 (21)/7 (50)
Hospitalization	
Duration (months): mean (±SD) [range]	10.8 (±5.5) [0.1–21.5]
Comorbidities: N (%)	
Syndromic ASD	7 (50)
Epilepsy	6 (43)
Catatonia	6 (43)
Obesity	3 (21)
OSA	1 (7)
Osteoporosis	1 (7)
GERD	3 (21)
Esophagitis, duodenitis, gastritis	11 (79)
Chronic constipation	12 (86)
ENT and maxillofacial infections	4 (29)
Parasitic infection	3 (21)
Cardiopathy	2 (14)
Anemia	3 (21)
Pruritic skin diseases	2 (14)
Iatrogeny	3 (21)
At baseline	
Patients receiving medication: N (%)	13 (93)
Poly-medication: N (%)	10 (71)
Equivalent of chlorpromazine (mg) per patient receiving medication: mean (±SD) [range]	664 (±665) [67–1750]
History of exception treatment: N (%)	
ECT	2 (14)
TBW	3 (21)
During the CG trial	
Change in medication: N (%)	2 (14)
Change in equivalent chlorpromazine	−105 mg in one patient, +150 mg in another patient
List of compounds: N (%)	10
Antipsychotic	6
Anticonvulsant	5
Antidepressant	3
Mood stabilizer	1
Psychostimulant	12
Melatonin	5
Antiparkinsonian	7
Benzodiazepine	
Gastrointestinal comfort treatment (laxative, gastric protector, anti-reflux drug)	26
Oxygen therapy	1
Analgesic (level 2 or 3)	3
Other drugs	3
Physiotherapy: N (%)	4
Autism History	
ADI-R: 4–5 years, mean (±SD) *	
Social impairment score	26.1 (±2.8)
Communication score	9.6 (±2.1)
Repetitive interests score	7.7 (±2.4)
Developmental score	4.7 (±1.8)
Clinical Characteristics	
Main reason for referral	Self-injurious behavior (N = 7) Catatonia (N = 2) Agitation (N = 2) Destructive behavior (N = 2) Hetero-aggression (N = 1)
Language	Fluent (N = 0) Few words (N = 7) Non-verbal (N = 7)
ABC, mean (±SD) [range]	61 (±23) [21–109]
CARS, mean (±SD) [range]	38 (±5) [30–51]
Vineland Adaptive Behavior Scales-II **	
Developmental age: years	2.7 (±1.1) [1.8–5]

* N = 10 due to missing data; ** N = 11 due to missing data; ABC, Aberrant Behavior Checklist; ADI-R, Autism Diagnostic Interview-Revised; ASD, autism spectrum disorder; CARS, Children Autism Rating Scale; CG, compression garment; ECT, electroconvulsive therapy; ENT, ear, nose, and throat; GERD, gastroesophageal reflux disease; mths, months; OSA, obstructive sleep apnea; SD, standard deviation; SES, socioeconomic status; TWB, therapeutic body wrap.

**Table 3 children-07-00077-t003:** Clinical changes in outcome variables between the baseline and T-1 (after CG) (N = 14).

Variable	Baseline	T-1 (after CG)	Delta	Effect Size	*p* *
ABC Scores					
ABC-irritability	21.64 (9.52)	16.46 (9.03)	−5.18 (7.78)	−0.65	0.028
ABC-lethargy	12.79 (8.14)	7.57(6.89)	−5.21 (4.25)	−1.19	0.002
ABC-stereotypies	7.64 (3.77)	6.71 (3.09)	−0.93 (2.5)	−0.36	NS
ABC-hyperactivity	17.36 (12.59)	11.61 (8.85)	−5.75 (7.97)	−0.7	0.003
ABC-inappropriate speech	1.79 (3.38)	1.68 (3.21)	−0.11 (1.76)	−0.06	NS
ABC-total	61.21 (23.44)	44.04 (20.19)	−17.18 (20.19)	−0.83	0.008
DUNN Questionnaire					
Auditory processing	2.2 (0.92)	2.4 (0.7)	0.2 (0.79)	0.24	0.586
Visual processing	2.6 (0.7)	2.7 (0.48)	0.1 (0.57)	0.17	0.773
Vestibular processing	2.5 (0.85)	2.6 (0.7)	0.1 (0.74)	0.13	1
Touch processing	2 (0.82)	2.3 (0.82)	0.3 (0.82)	0.35	0.345
Multisensory processing	1.9 (0.74)	2.1 (0.88)	0.2 (0.92)	0.21	0.572
Oral sensory processing	2.2 (0.92)	2.4 (0.84)	0.2 (0.79)	0.24	0.586
Sensory processing related to endurance/tone	2 (0.82)	2 (0.82)	0 (0.47)	0	1
Modulation related to body position and movement	2.1 (0.88)	2.3 (0.82)	0.2 (0.63)	0.3	1
Modulation of movement affecting activity level	2.9 (0.32)	3 (0)	0.1 (0.32)	0.3	1
Modulation of sensory input affecting emotional responses and activity level	2.5 (0.71)	2.5 (0.71)	0 (0.82)	0	1
Modulation of visual input affecting emotional responses and activity level	2 (0.47)	2.4 (0.7)	0.4 (0.52)	0.74	0.072
Emotional/social responses	1.7 (0.95)	1.7 (0.67)	0 (0.94)	0	1
Behavioral outcomes of sensory processing	2 (0.82)	1.8 (0.63)	−0.2 (0.42)	−0.45	0.346
Items indicating thresholds for responses	2.1 (0.88)	2.3 (0.95)	0.2 (0.42)	0.45	0.346
DUNN Factors					
Sensory seeking	2.7 (0.67)	2.8 (0.63)	0.1 (0.32)	0.3	1
Emotionally reactive	2.1 (0.88)	2.1 (0.88)	0 (0.67)	0	1
Low endurance/tone	2 (0.82)	2.1 (0.88)	0.1 (0.57)	0.17	0.773
Oral sensory sensitivity	2.5 (0.71)	2.5 (0.85)	0 (0.82)	0	1
Inattention/distractibility	2.5 (0.71)	2.6 (0.52)	0.1 (0.74)	0.13	1
Poor registration	1.6 (0.7)	2.1 (0.88)	0.5 (0.53)	0.91	0.037
Sensory sensitivity	2.3 (0.95)	2.5 (0.85)	0.2 (0.42)	0.45	0.346
Sedentary	2.8 (0.42)	2.7 (0.48)	−0.1 (0.32)	−0.3	1
Fine motor/perceptual	1.9 (0.99)	1.5 (0.85)	−0.4 (0.7)	−0.55	0.174
Motricity Path					
Successful items	23.73 (18.45)	29.91 (20.43)	6.18 (7.31)	0.81	0.025
Emerging items	12.91 (9.88)	11.82 (9.9)	−1.09 (6.5)	−0.16	0.442
Failed items	13.27 (12.19)	8.18 (8.87)	−5.09 (9.16)	−0.53	0.074
Postural Control					
Frontal (mediolateral)	NA	NA	0.92 (0.67)	1.37 **	0.006
Profile (anteroposterior)	NA	NA	1.36 (0.81)	1.69 **	0.007

* Wilcoxon signed rank test; ** Cohen’s d; ABC, Aberrant Behavior Checklist; CG, compression garments; NA: not appropriate.

**Table 4 children-07-00077-t004:** Postural changes after wearing CG at six weeks (N = 14).

	hEDS/ HSD *	Residual Symptoms before CG	Profile View			Dorsal View				CGI after CG
			Post Scapular Plan dev	Ant Scapular Plan dev	Changes Observed with CG	Global Lateral Deviation	Cervico-Dorsal Deviation	Lateral Rotation Deviation	Changes Observed with CG	
P1	yes	SIB	1	0	no change	1	0	0	improved	improved
P2	yes	SIB	0	1	no change			0	normalized	improved
P3	yes	SIB	0	1	improved	0	0	1	improved	improved
P4	yes	SIB	0	1	improved	1	1	0	improved	improved
P5	yes	catatonia	0	1	normalized	0	1	0	improved	improved
P6	yes	SIB	0	0	no change	1	0	0	improved	improved
P7	yes	SIB	NE	NE	NE	NE	NE	NE	NE	improved
P8	no	SIB	NE	NE	NE	NE	NE	NE	NE	worsen
P9	no	instability	0	0	no change	1	0	0	normalized	improved
P10	no	SIB	NE	NE	NE	1	0	0	normalized	improved
P11	no	catatonia	1	0	normalized			0	no change	improved
P12	no	SIB	0	1	normalized	1		0	no change	improved
P13	no	instability	0	1	normalized	1	0	0	no change	improved
P14	no	hyperpnea	0	1	normalized	1	0	0	no change	improved

* based on the criteria of Malfait et al., 2017 [43]; 0, absent; 1, present; CG, compression garments; hEDS, hypermobility Ehlers–Danlos syndrome; HSD, hypermobility spectrum disorder; NE, not evaluable; SIB, self-injurious behaviors; post, posterior; ant, anterior; dev, deviation; CGI, clinical global impression.

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
