# Peer review of "Compressive Garments in Individuals with Autism and Severe Proprioceptive Dysfunction: A Retrospective Exploratory Case Series"

_children, 2020, doi:10.3390/children7070077_

Round 1

Reviewer 1 Report

The paper is overall well written and has been pleasant to read. The aim of the manuscript is to explore the use of Compression garments (CGs) in individuals with autism and severe proprioceptive dysfunction (SPD), including individuals with Generalized joint hypermobility (GJH), to control posture and challenging behaviors. Orthodox interventions are not completely effective in some patients because they leave residual symptoms. These residual symptoms may be related to alterations in sensory processing, including changes in the integration of information across the different sensory modalities .

Main limitation: External review board should be required although it is a retrospective design. Also, inscription in clinical trial web is necessary for many scientific journals.

The sample size was limited and the study retrospective with no blind clinical assessments.

Author Response

Reviewer 1

The paper is overall well written and has been pleasant to read. The aim of the manuscript is to explore the use of Compression garments (CGs) in individuals with autism and severe proprioceptive dysfunction (SPD), including individuals with Generalized joint hypermobility (GJH), to control posture and challenging behaviors. Orthodox interventions are not completely effective in some patients because they leave residual symptoms. These residual symptoms may be related to alterations in sensory processing, including changes in the integration of information across the different sensory modalities.

We thank reviewer 1 for this encouraging remarks.

Point 1. Main limitation: External review board should be required although it is a retrospective design. Also, inscription in clinical trial web is necessary for many scientific journals

We understand reviewer’s remark. However, local rules need to be followed as well. Compared to Foreign rules (local ethical committee), in France, we have 3 levels of ethical requests that are separated and not under the umbrella of a unique committee.

  1. Medical data bases have very strict rules under a national authority (Commission Nationale Informatique et Liberté). This commission exists since more than 20 year and has promoted the RGPD rules in Europe that has recently been adopted by EU.
  2. Local Research Ethical Committee for experimental research with low risk
  3. Commité de Protection des Personnes (Committee for Patients’ Protection, CPP) for experimental studies with potential risk. In that case it is forbidden to request a local committee and you must registered to a National Agency registration and an independent CPP is randomly chosen outside the local trust coordinating the research.

As stated in the method section, for retrospective studies we are in case 1. To answer reviewer querie we revised the paragraph and added the registration number of our computational data base in the section Materials and Methods. It is now P 3, lines 134-136: “As we employed a retrospective design for this case series, no external review board was required [53]. However, given the compassionate nature of the intervention, written informed consent was obtained from the parents of all participants when CG were proposed. In addition, as requested by French regulation rules, all patients/families were contacted for inclusion of the data in a retrospective study; in particular, the database was approved by the Commission Nationale Informatique et Liberté (CNIL or National Informatics and Freedom Commission) under the number N°1256552.

Point 2. The sample size was limited and the study retrospective with no blind clinical assessments.

We thank reviewer 1 for this remark that we share. We changed the limitation section as follow. It is now in the discussion, page 17, line 376: “The study has several important limitations: First, the clinical sample was biased towards the most severe and resistant cases. Second, the sample was heterogeneous, as proprioceptive dysfunction was associated or not with joint hypermobility or hEDS. Third, the sample size was limited and the study was retrospective with no historical control group and no blind clinical assessments. Fourth, […].”

Reviewer 2 Report

-Please define “proprioceptive.”  This is a term that may confuse unfamiliar readers.

- “Some maladaptive behaviors associated 87 with weak proprioceptive integration are identifiable in clinical practice through observations 88 [31-34]”  A brief explanation of some of these findings would hep to provide context for the arguments that the authors are making.

-Just to be clear, a brief discussion of the ages of participants should be provided.  Given that this article was submitted to Children, it should be made clear at least once that the studied population does indeed consist of children.

-Page 8, line 361, “Forth” should be “Fourth.”

-While the exploratory nature of the study and the methodological limitations are acknowledged, additionally acknowledging that there was no control group is necessary.  Further, the reasons why this is a major limitation should be discussed in their entirety so the reader understands the major issues with relying only on pre/post test designs and making bold claims about potential relevance of results.  This really tempers the findings of the study in my opinion.

Author Response

Point 1. Please define “proprioceptive.”  This is a term that may confuse unfamiliar readers.

As requested we added a definition of proprioception in the introduction section. It is now: We added, P 2, lines 76-79: “Proprioception is the sensory registration of the ongoing spatial configuration of the body, which includes the position of body segments in space, the force and speed of movement, and the integration of gravity and body balance.

Point 2.  “Some maladaptive behaviors associated with weak proprioceptive integration are identifiable in clinical practice through observations [31-34]”  A brief explanation of some of these findings would help to provide context for the arguments that the authors are making.

We thank reviewer 2 for this request that should help readers unfamiliar with severe cases to better understand our meaning. In the revised version of the MS, we added, P 2, lines 90-92: “Some maladaptive behaviors associated with weak proprioceptive integration are identifiable in clinical practice through observations: postural maladjustments (e.g., tonic postural collapse with unusual points of gravitational support, such as the hands or the head), sensory-motor stimulations (e.g., twirling), and unsteady balance [31-34]”. 

Point 3. Just to be clear, a brief discussion of the ages of participants should be provided.  Given that this article was submitted to Children, it should be made clear at least once that the studied population does indeed consist of children.

We thank reviewer 2 for this remark. This is legitimate given the readership of Children. As requested we make the point of age very clear in the result section. It is now page 8, line 254 of the revised version of the MS: “The study sample included 14 participants: 13 males and 1 female. Our cohort consisted of 7 children and adolescents (aged 8–15 years) and 7 young adults (aged 18–30 years). All of them had severe autism, no functional language, and a mean developmental age of less than 36 months with respect to the Vineland Adaptive Behavior Scale II (VABS-II).

We also added in table 2 the Vineland Adaptive Behavior Scales-II developmental ages.

However, many adults with severe autism and ID are not treated in adult psychiatrists but rather by child psychiatrists given the developmental ages (here less than 36 months).

Point 4. Page 8, line 361, “Forth” should be “Fourth.”

Thanks for noticing this spelling error. Change was made accordingly.

Point 5. While the exploratory nature of the study and the methodological limitations are acknowledged, additionally acknowledging that there was no control group is necessary.  Further, the reasons why this is a major limitation should be discussed in their entirety so the reader understands the major issues with relying only on pre/post test designs and making bold claims about potential relevance of results.  This really tempers the findings of the study in my opinion.

We thank reviewer 2 for this remark that was shared also by reviewer 1. We changed the limitation section accordingly. It is now in the revised MS: “The study has several important limitations: First, the clinical sample was biased towards the most severe and resistant cases. Second, the sample was heterogeneous, as proprioceptive dysfunction was associated or not with joint hypermobility or hEDS. Third, the sample size was limited and the study was retrospective with no historical control group and no blind clinical assessments. Fourth, […].”

Also in the paragraph ‘Implications for future research’, we added, P 17, line 410 “These promising results encourage the design of a randomized and controlled multi-center protocol, in order to assess not only sensory-motor skills but also the core symptoms of autism”.

Round 2

Reviewer 1 Report

I consider that although the limitations are better explained in the current version, they prevent its publication. I still consider that the sample is insufficient, as a pilot study, for the quality of the journal.

Author Response

Authors thanks reviewer 1 for his/her careful comments on our MS. We are happy that all the revisions we made were satisfactory. It remains however one important concern that is the sample size.

As we present a retrospective exploratory study, we can't change the sample size at this stage. As we stated, our goal is to organize a RCT based on this first publication. The population we had in this study is among the most severe in the field of ASD and ID. So I believe that an exploratory study would be warranted by any ethical committee to proceed on a RCT. 

Regarding the sample size, I asked our statistical expert to calculate the power of the results we present in the MS. Detailed justification is given in the uploaded document regarding the two tests we used (Wilcoxon and Friedman). As you can see power (Beta error) was not below 0.8 which is sufficient for an exploratory study.
